# Estrogen Receptor Alpha Splice Variants, Post-Translational Modifications, and Their Physiological Functions

**DOI:** 10.3390/cells12060895

**Published:** 2023-03-14

**Authors:** Kenji Saito, Huxing Cui

**Affiliations:** 1Department of Neuroscience and Pharmacology, Carver College of Medicine, The University of Iowa, Iowa City, IA 52241, USA; huxing-cui@uiowa.edu; 2Iowa Neuroscience Institute, Carver College of Medicine, The University of Iowa, Iowa City, IA 52241, USA; 3F.O.E. Diabetes Research Center, Carver College of Medicine, The University of Iowa, Iowa City, IA 52241, USA

**Keywords:** estrogen receptor alpha, splicing isoforms, post-translational modifications, mutant mouse models

## Abstract

The importance of estrogenic signaling for a broad spectrum of biological processes, including reproduction, cancer development, energy metabolism, memory and learning, and so on, has been well documented. Among reported estrogen receptors, estrogen receptor alpha (ERα) has been known to be a major mediator of cellular estrogenic signaling. Accumulating evidence has shown that the regulations of ERα gene transcription, splicing, and expression across the tissues are highly complex. The ERα promoter region is composed of multiple leader exons and 5′-untranslated region (5′-UTR) exons. Differential splicing results in multiple ERα proteins with different molecular weights and functional domains. Furthermore, various post-translational modifications (PTMs) further impact ERα cellular localization, ligand affinity, and therefore functionality. These splicing isoforms and PTMs are differentially expressed in a tissue-specific manner, mediate certain aspects of ERα signaling, and may work even antagonistically against the full-length ERα. The fundamental understanding of the ERα splicing isoforms in normal physiology is limited and association studies of the splicing isoforms and the PTMs are scarce. This review aims to summarize the functional diversity of these ERα variants and the PTMs in normal physiological processes, particularly as studied in transgenic mouse models.

## 1. Estrogen Receptor Alpha (ERα)

Estrogens are steroid sex hormones that are critical for the development of reproductive organs and reproductive functions primarily in females. In addition to reproduction, estrogenic signaling also plays important roles in cardiovascular physiology, homeostatic regulation of energy balance, and a variety of social and learning behaviors. Traditionally, the actions of circulating estrogen were believed to be mediated mainly by binding to two specific receptors, estrogen receptors α (ERα) and estrogen receptors β (ERβ), in which liganded ERs activate gene transcription through binding to the genomic element called the estrogen-response element (ERE), either as a homodimer or heterodimer with coactivators. Notably, apart from their well-known roles in transcriptional regulation, estrogens also mediate rapid signaling cascades called membrane-initiated pathways, a new mode of action via membrane ERs and a G-protein-coupled estrogen receptor (GPER, also known as GPR30).

ERα and ERβ belong to the nuclear receptor superfamily of ligand-inducible transcription factors, which also contains the receptors for other steroid hormones including glucocorticoids, mineralocorticoids, progestogens, androgens, retinoic acid and thyroid hormone. The structure of the nuclear receptor is characteristic and well conserved among the subfamilies, consisting of a variable N-terminal ligand-independent transactivation domain (activation function-1, AF-1), a DNA-binding domain (DBD), a hinge domain and a C-terminal ligand-binding domain (LBD) and a ligand-dependent transactivation domain (AF-2) (Figure 1).

A number of ERα splicing variants have been described without a corresponding understanding of the physiological functions of each of those isoforms. Alternative splicing is an important molecular mechanism that enables the production of a variety of different transcripts from a single gene by selectively arranging coding exons from pre-mature mRNA. Many of them are reported particularly in cancers and tumor cells, but some of them are also expressed constitutively in normal tissues. In this review, we will primarily focus on the identified functional diversity of ERα variants in physiological conditions. For the reviews on an extensive list of ERα variants in cancer, we refer the readers to previous excellent reviews [1,2,3,4].

**Figure 1 cells-12-00895-f001:**
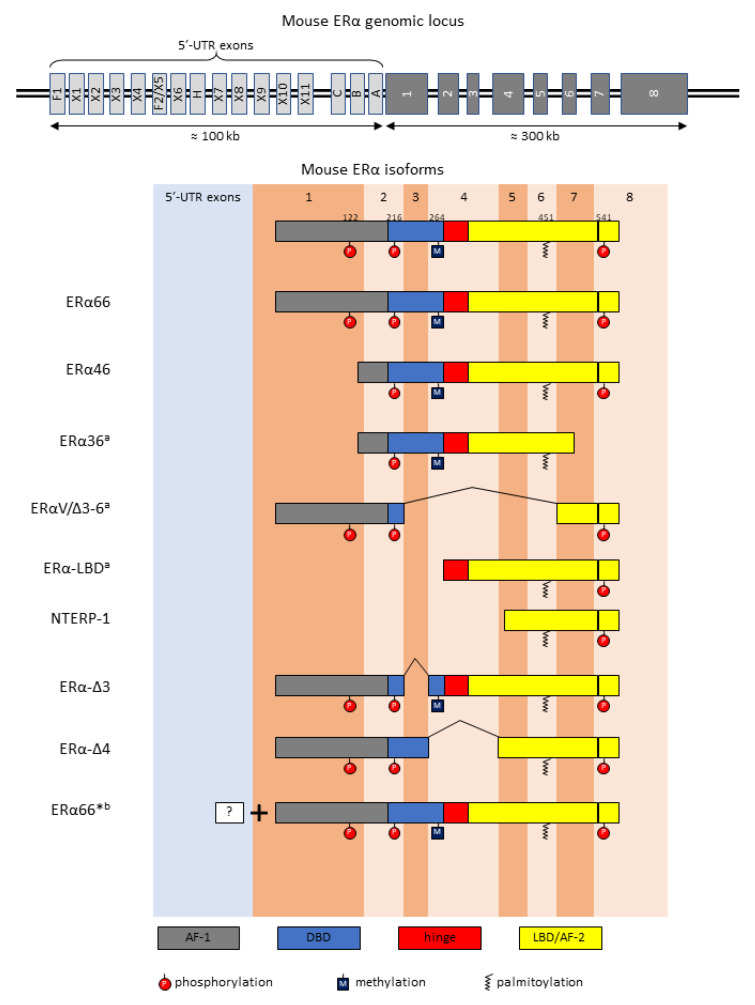
Mouse ERα gene organization, alternative splicing isoforms, and post-translational modifications discussed in the review. Mouse ERα gene organization is based on Ishii et al. and Kos et al. [5,6]. The illustration does not reflect actual lengths of exons and introns. ERα functional domains are aligned along their coding exons. Exon numbering is based on full-length ERα or ERα66. The PTMs are displayed if the amino acid residues are preserved in each isoform regardless of experimental proof. AF-1/2; activation function-1 and 2, DBD; DNA-binding domain, hinge; hinge domain, LBD; ligand-binding domain. ^a^ ERα36, ERαV/Δ3-6, and ERα-LBD are not fully characterized in mice. ^b^ Alternative splicing of ERα66* is not fully characterized.

## 2. Complex Organization of ERα 5′-UTR

Alternative promoter usage enables the generation of multiple mRNA species from a single gene. ERα genes across species shares highly complex organization in the promoter region. At least four leader exons and 12 internal exons in mice [5,6] (Figure 1); four leader exons and 11 internal exons in rats [7]; and seven leader exons and four internal exons in humans [6,8] have been reported, and the list of 5′-UTR exons is growing. Alternative usages of leader exons and 5′-UTR internal exons lead to a wide variation of mRNA transcripts. Interestingly, although many mouse ERα transcripts with various 5′-UTR sequences are identified and their expressions are regulated in a tissue-specific manner, many of them produce the same ERα protein (full-length ERα or ERα66) [5,6]. Each cell type has a differential availability of transcription factors, co-transcription factors, and regulatory proteins; and transcription factor response elements spread throughout the ERα promoter region. Although the biological significance of the production of identical ERα proteins from multiple transcripts is unknown, multiple promoter usages can achieve context-dependent regulation of ERα expression and therefore robustness of ERα functions.

## 3. ERα Isoform Variation and Functions

In addition to full-length ERα or ERα66, a number of ERα splice isoforms have been reported in normal tissues and pathological conditions (typically in tumors), and they are often co-expressed with full-length ERα. In this review, we prioritize ERα variants that are expressed in normal physiological conditions. In this chapter, we discuss their physiological functions primarily derived from in vitro studies, but put emphasis on knowledge from in vivo studies. However, since splice variants and full-length ERα share the same exons as each other, there is an extremely limited number of studies to tease out splice variant-specific physiological functions. Furthermore, due to limited access to human tissues, the majority of in vivo observation comes from rodent studies. Although little is known about how the expression of ERα splicing variants is regulated, we include a discussion of the regulatory mechanisms when available. Figure 1 summarizes ERα splicing variants and the PTMs discussed in this review. Our current understanding of their properties and physiological functions is summarized in Table 1.

### 3.1. Differential Splicing

#### 3.1.1. ERα46

A 46-kDa isoform lacking first 173 N-terminal amino acids or AF-1 was first found in MCF-7 mutant human breast cancer cell lines and named “ERα46” for its molecular weight 46 kDa, which shares an identical DNA sequence with ERα66 for the remaining portion [9]. Sequencing of the ERα46 transcript revealed that ERα46 is translated from an ATG located in the full-length ERα’s exon 2 (Figure 1). Its transcription may be associated with usages of particular 5′-UTR exons. Flouriot et al. have characterized six 5′-UTR exons and have showed that the transcription of ERα46 is initiated from either exon 1E or F and spliced to exon 2E, which is preferentially spliced to exon 2 to produce ERα46 mRNA [9]. In addition, proteolysis or use of alternative translation initiation is also suggested to produce ERα46 [10].

Two molecules have been reported to be involved in the regulation of ERα46 expression at the transcription and splicing levels. Ohe et al. have shown that the high-mobility group A protein 1a (HMGA1a) is involved in the alternative splicing of ERα to induce ERα46 expression in MCF-7 mammary carcinoma cells [37]. HMGA1a is known as an oncogene and its expression is correlated with tumor malignancy. The authors found that the HMGA1a RNA-binding sequence is located at the ERα exon 1. Transient expression of HMGA1a in MCF-7 cells induces ERα46 expression associated with reduced expression of ERα66, which is blocked by the HMGA1a RNA decoy competing with the ERα transcript for HMGA1a binding, suggesting that HMGA1a regulates alternative splicing of ERα in favor of ERα46 expression. A homeobox transcription factor BARX2 binding site is located upstream of a 5′-UTR exon 2E and the overexpression of BARX2 in MCF-7 cells doubles the direct splicing from exon 2E to exon 2, suggesting that BARX2 involves in the transcription and splicing of the ERα46 [38].

ERα46 expression was confirmed in normal human tissues including endometrium, ovary, liver, skeletal muscles, lung, kidney, and adrenal gland, but not the pituitary [9]. The rat counterpart is also expressed in various tissues [12]. Irsik et al. showed that ERα46 is most abundant in the heart of both sexes; the female uterus, and male testis [13]. They also studied the subcellular distribution of ERα46 in the renal cortex and found that ERα46 was primarily located at the membrane fraction, suggesting that ERα46 may be involved in membrane-initiated estrogenic signaling [13]. When ERα46 is transfected in cultured cells, they can be located either in the nuclei [12] or both in the nuclei and extranuclei [11]. When ERα46 and ERα66 are co-expressed in cultured cells, ERα46 powerfully inhibits the AF-1 activity of ERα66 [9].

#### 3.1.2. ERα36

ERα36 was found in MCF-7 breast cancer cells, which lacks both activation function domains, retains the DNA binding domain and a part of the ligand binding domain, and has a unique addition of 27 amino acids on the C-terminus encoded by a previously unknown exon [14]. This variant has been primarily studied in cancer cells and its expression in normal human tissues is not fully characterized. However, ERα36 expression has been reported at least in normal endometrial tissues with significantly higher levels than endometrial carcinoma tissues [39], gastric tissues with lower levels than that of tumor tissues [40], colon [41], and osteoblast [42]. The last exon coding the unique 27 amino acids does not exist in mice, and the mouse counterpart of human ERα36 is not clearly identified. However, the immunoreactivity using a human ERα36-specific antibody was observed in mouse kidney [13], and ovary and oocytes [43]. Multiple reports also detect a 36-kDa band on immunoblots from rodent tissues including brain, kidney, liver, heart, lung, mammary gland, ovary, uterus, and testis [13,36,43,44]. Taken together, the mouse counterpart of human ERα36 is likely expressed in normal mouse tissues.

At the subcellular level, ERα36 is located at the plasma membrane [13,18]. ERα36 is speculated not to activate transcription due to the lack of two AF domains, and to antagonize transcriptional activity by other ERs based on the retention of a DBD and a partial LBD [45]. Furthermore, its distribution on plasma membrane may imply that this variant primarily works in membrane-initiated pathways. Thus far, the physiological roles of ERα36 are largely unknown. Thiebaut and colleagues have developed mouse models in which ERα36 is overexpressed in the mammary gland [16] or the male germ cells [17]. The overexpression of ERα36 in the mammary gland caused constitutive estrogenic activation phenotypes including hyperdilation of the duct, lumen thickening, epithelium thinning, and leakage in the mammary gland [16]. On the other hand, while the overexpression of ERα36 in the male germ cells altered gene expression in the cell proliferation, meiotic division, and differentiation, no defects in cell differentiation, sperm quality, or fertility have been observed in adult transgenic male mice [17].

#### 3.1.3. ERαV/ERαΔ3-6

Ohshiro et al. reported that a nuclear receptor-box-containing full-length protein, called nuclear protein E3-3 (NPE3-3), binds to ERα [20]. When this protein is overexpressed in MCF-7 cells, they observed a reduction in ERα66 expression as well as the appearance of a new 37-kDa protein band, which the authors identified as a novel ERα isoform ERαV that is composed of exons 1, 2, 7, and 8. Thus, ERαV lacks large portions of DNA- and ligand-binding domains that are encoded by exons 3–6 (Figure 1). Since NPE3-3 interacts with splicing factors, this study suggests that NPE3-3 acts as a trans-acting splicing regulatory factor and gathers splicing factors to induce alternative splicing of the ERα transcript to produce ERαV [20]. Ishunina and Swaab reported an ERα variant of the same exon composition as ERαV in human brain nuclei (ERαΔ3-6) [46], indicating that the ERαV variant is expressed in normal tissues.

The E_2_-dependent transcriptional activity of ERαΔ3-6 is significantly lower than that of wild-type (WT) ERα [19]. The expression of ERαΔ3-6 is modified by estrogens in a cell type-specific manner; estradiol administration enhances the expression of ERαΔ3-6 in M17 human neuroblastoma cell lines but not in Hela human cervical cancer cell lines [19].

One must be careful when ERαV or ERαΔ3-6 is studied in Western blot (WB) because its molecular weight is close to those of ERα36 and ERα-LBD, mentioned next. These two splice isoforms have to be carefully characterized in WB analysis, depending on the ERα antibodies, against different epitopes. Currently, there is no report for the identification of the rodent counterpart of ERαV. However, ERα WB studies often reported bands below 40 kDa by antibodies against either the ERα N-terminus or C-terminus that is missing from ERα36, and they are expressed differentially in multiple tissues and in different subcellular fractions [13,36,47].

#### 3.1.4. ERα-LBD

Recently, Strillacci et al. reported an ERα isoform lacking the N-terminal domains including AF-1, DBD, and a part of hinge region, in breast cancer cell lines; they named this ERα-LBD because it is composed of LBD and AF-2 [21]. The transcript is suggested to utilize an alternative transcription start site on the 5′ upstream of exon 4, which is spliced to exon 4–8. The molecular weight of ERα-LBD is 37.3 kDa, which is close to those of ERα36 and ERαV. However, since each possesses unique amino acid sequences, the ERα36-specific antibody did not detect the ERα-LBD. ERα-LBD was largely found in the cytoplasm and mitochondria of breast cancer cell lines. Overexpression of ERα-LBD promoted breast cancer growth and made cancer cells resistant to ERα antagonist treatment. Authors searched public RNA-seq data but did not find ERα-LBD expression in normal breast tissues. Therefore, it is unclear whether this isoform is expressed in other normal tissues and has physiological functions. An ERα transcript initiated from the 5′ upstream sequence of exon 4, which is supposed to produce an identical protein of ERα-LBD, has been reported in human uterine endometrial tissue from uterine leiomyoma patients [48].

#### 3.1.5. N-Terminally Truncated ERα Products (NTERPs)/Truncated Estrogen Receptor Products (TERPs)

N-terminally truncated ERα products, NTERPs, which encode splicing variants lacking the N-terminal domain through the hinge domain, have been reported in mouse reproductive organs, brain, kidney, and pituitary [22]. The endogenous expression assay revealed that they are extranuclear proteins and E_2_ did not change their subcellular localization [22].

TERP-1 and 2 were found in female rat pituitary [23]. Their protein products are the same as those of NTERPs in mice. The transcription of TERPs starts from the end of exon 4, and one is directly spliced to exon 5 (TERP-1) while the other retains part of the intronic sequence and connects to exon 5 (TERP-2). The protein products are 20–24 kDa in molecular weight and retain part of the LBD and AF-2 [23]. Estradiol treatment potently induced the expression of TERP-1/2 in male rat pituitary [23]. In fact, the TERP-1 promoter located between exons 4 and 5 was activated by estradiol in GH3 somatolactotrope cells transiently transfected with the TERP-1 promoter reporter construct [24,25]. TERP-1 works as an inhibitor of ERα in transfected CHO cell lines; it inhibited ERE-binding of ERα, E_2_-ERα interaction, E_2_-induced transactivation, and nuclear localization of ERα [24]. Mouse TERP-1 has also been reported [22,26]. The TERP-1 is expressed in ERα-Neo knockout (KO) mice, in which the majority of AF-1 domain is deleted.

#### 3.1.6. ERαΔ3, ERαΔ4, ERαΔ3/4

In the rat pituitary, three splicing isoforms are reported to be differentially expressed during development [28]. The authors detected ERα isoforms lacking exon 3 (ΔE3), exon 4 (ΔE4), and exon 3/4 (ΔE3-4), as well as full-length ERα; and predicted their molecular weights as 61.8, 53, 45, and 67 kDa, respectively. During the embryonic development, ΔE4 and ΔE3-4 are predominant; and ΔE3 is abundant transiently during the late gestation, while the full-length ERα increases dramatically from embryonic day 19 and becomes dominant after birth throughout the lifetime. Their follow-up study showed that ΔE3 inhibits estradiol-dependent transcription by full-length ERα in co-transfected CHO cells, while ΔE4 showed ligand-independent transcription [27]. ERαΔ4 has been reported in immortalized mouse hypothalamic neurons [30,34] and mouse brain [35], and predominantly located in the cell membrane [29]. While the human ERαΔ4 transcript is reported in various cell lines and tissues including MCF-7 cells [31] and human pituitary [32], the protein expression is confirmed in ovarian carcinoma [33].

#### 3.1.7. ERα-Neo KO Mouse

Since ERα splicing isoforms share DNA sequence with full-length ERα, the deletion of any exons of the isoforms will also affect the expressions of other ERα isoforms including full-length ERα. However, there is a mouse model in which ERα AF-1 is deleted. Thus, in this mouse model, the expression of full-length ERα is disrupted but those of other ERα isoforms including ERα46 are maintained. Lubahn et al. first reported ERα gene knockout in mice by replacing the majority of first coding exons with a Neomycin-resistance gene; thereafter it is called ERα-Neo KO mice [49]. Later, this mouse line turned out to possess ERα46 and a chimeric 55-kDa ERα with a partial Neo cassette (ERα55) [50,51]. Thus, the physiological roles of ERα isoforms other than full-length ERα could be speculated by comparing the phenotypes of ERα-Neo KO and other complete ERα KO mice. E_2_-dependent uterine development and the expression of ERα target genes are not fully blocked in ERα-Neo KO mice [51,52], indicating that full-length ERα as well as the splice isoforms are required for normal functions of the uterus. The E_2_ effect on endothelial NO production was maintained in the ERα-Neo KO mice while such actions were abolished in complete ERα KO mice [51]. The E_2_-dependent response against vascular injury was maintained in ERα-Neo KO mice [53], while such effects were dampened in complete ERα KO mice [54,55]. These studies indicate that the estrogenic effects on the vasculature are likely to be mediated primarily by the full-length ERα.

ERα alternative splicing variant expressions show distinct temporal dynamics, tissue-specific patterns, and developmental changes. The regulatory mechanisms of ERα alternative splicing have been largely unknown except for a few examples (HMGA1 and BARX2 for ERα46 and NPE3-3 for ERαV). In addition to understanding the distinct physiological functions of those isoforms, it is important to analyze the mechanisms behind these differential expression profiles.

### 3.2. Post-Translational Modifications

In addition to differential splicing, PTMs of ERα add another layer of functional diversity in ERα biology. Multiple phosphorylation sites and other PTM sites have been reported in ERα proteins [56]. In this chapter, we briefly discuss ERα PTMs whose physiological functions have been evaluated in vivo primarily using genetic mouse models (Table 2). For a more comprehensive description of ERα PTMs, the readers are referred to other excellent reviews [56,57,58].

#### 3.2.1. S118 Phosphorylation

Serine 118 (S118) is the most widely studied phosphorylation site of human ERα that is located in the AF-1 domain. S118 phosphorylation is important for the dimerization of ERα and the induction/suppression of several ERα target genes [59,60]. S118 phosphorylation facilitates its interaction with some co-activators and then mediates the ligand-dependent/independent activation of ERα [61,62]. Some hormones and growth factors, such as prolactin, epidermal growth factor, and insulin-like growth factor-1 can induce S118 phosphorylation [63,64].

**Table 2 cells-12-00895-t002:** Post-translational modifications of ERα and the physiological functions.

Site in Human ERα	Site in Mouse ERα	Modification	Physiological Functions	References
S118	S122	Phosphorylation	Dimerization of ERα/transcriptional regulation	[59,60]
			Sexually dimorphic effects on fat accumulation; amino acid substitution to non-phosphorylatable form leads to more fat accumulation in female mice and lower fat mass in male mice	[65,66]
S212	S216	Phosphorylation	Transcriptional regulation via ERE	[67]
			Amino acid substitution to non-phosphorylatable form increases inflammatory signatures and impairs motor coordination	[68]
R260	R264	Methylation	ERα cytoplasmic distribution	[69]
			Amino acid substitution to non-methylatable form impairs membrane-initiated actions of E_2_ in the endothelial cells	[70]
C447	C451	Palmitoylation	ERα plasma membrane localization	[71,72]
			Amino acid substitution to non-palmitoylable form leads to female mouse infertility and impairs the homeostatic regulation of trabecular bone in the axial skeleton	[73,74,75]
Y537	Y541	Phosphorylation	E_2_-ERα binding/interaction with kinases	[76,77]
			Amino acid substitution to non-phosphorylatable form leads to female mouse infertility	[78]

The mouse counterpart of human ERα S118 is S122. A genetic mouse model harboring the amino acid substitution of serine to alanine at the 122 (S122A) of ERα has been generated so that site 122 is no longer phosphorylated. Mutant S122A ERα female mice show normal uterine weight, normal expressions of estrogen-regulated genes, normal estrogen-induced phosphorylation of Akt, and normal serum levels of estradiol and testosterone [65]. ERα signaling is involved in the regulation of bone development and maintenance [79,80,81]. Postmenopausal women often experience bone loss and increased risk of osteoporosis [82]. The mutant S122A ERα mice also display normal mineral density and skeletal architecture of the bones in both females and males [65,66]. The mutant S122A ERα mice show sexually dimorphic tissue-specific genetic effects. The mutant female mice show significantly more fat accumulation, increased adipocyte size, and body weight gain associated with an increase in fat mass composition [65]. On the other hand, while mutant S122A ERα male mice show normal sex steroid hormone levels, they show significantly lower body weight associated with lower fat mass after 9 months of age [66]. Since estrogens themselves facilitate S118 phosphorylation of human ERα [63], this sex differences of genetic effects may be due to the ligand abundance in females. To the best of our knowledge, sex difference of phosphorylation status of this site is not extensively evaluated in either humans or rodents.

#### 3.2.2. S212 Phosphorylation

Serine at the 212 located in the ERα DBD is also a target of phosphorylation. The phosphorylation status of this site determines the transactivation of the distinct group of genes via ERE when they are overexpressed in Huh-7 cells [67]. This serine 212 is phosphorylated in neutrophils infiltrating the uterus [83].

This serine residue of human ERα is conserved in mouse ERα at serine 216. A mutant mouse harboring an amino acid substitution of the non-phosphorylated form (S216A) has been generated, and was analyzed for inflammatory regulation of microglia [68]. The S216 of ERα is constitutively phosphorylated in microglia but not in astrocytes. Microglia from S216A ERα mutant mice displays its activation, for example, as swelled, shortened, and thickened morphology. Pro-inflammatory genes were increased, while anti-inflammatory genes were reduced, in response to lipopolysaccharide treatment. These results indicate that the phosphorylation of S216 regulates microglial anti-inflammatory functions. Behavioral phenotypes were not fully evaluated, but both male and female mutant mice gained more weight and showed impaired motor coordination [68].

#### 3.2.3. R260 Methylation

Arginine at the 260, located in the ERα DBD, is a methylation target. The methylated ERα is localized exclusively in the cytoplasm, and this methylation is required for some of the extranuclear functions of ERα such as the activation of PI3K and Src pathways [56,69].

The mouse counterpart of the R260 of human ERα is R264. A genetic mouse model with an amino acid substitution from arginine (R) to alanine (A) has been generated so that this site is no longer methylated. R264A ERα mutant mice show normal bone physiology, normal sex hormone levels, and no body weight phenotype [84]. Although R264 in mouse ERα is located in the DBD, E_2_-regulated gene transcription is largely preserved in the R264A ERα mutant uterus, while the mutation disturbed the membrane ERα-mediated actions of estradiol such as the rapid dilation of mesenteric arteries and the acceleration of endothelial repair of carotid [70]. Recently, it was shown that ERα can work as an RNA-binding protein (RBP), and the RNA-binding domain (RBD) is located in amino acids 255–272 of human ERα [85]. This region is on the border of the DBD and the hinge domain, and R260 is within this region. RBPs are a main group of arginine-methylated proteins, and the arginine methylation of RBPs affects RBP-RNA interactions [86]. Although it is unclear how the methylation status of the ERα R260 (and R264 in mouse ERα) influences ERα’s function as an RBP, the substitution of four amino acids including R260 to unmethylated alanine disrupts the ERα’s RBP function, leading to changes in an alternative splicing and translation of the target genes without affecting the DNA-binding ability [85]. Thus, part of the genetic effects observed in R264A ERα mutant mice may reflect ERα functions as an RBP.

#### 3.2.4. C447 Palmitoylation

In addition to genomic ERα signaling pathway via direct/indirect interactions with genome, estrogens activate rapid, membrane-initiated ERα signaling cascades. Plasma membrane-bound ERα is palmitoylated at the cysteine 447 residue (C447) [71,72]. The C447 is known to be the sole palmitoylation site in ERα [57,87], and C447 palmitoylation is essential for ERα plasma membrane localization, in part through the interaction with caveolin-1 [88]. A point mutation of C447 to the non-palmitoylable form is sufficient to inhibit its interaction with caveolin-1 and ERα membrane-initiated rapid signaling [71].

The mouse counterpart of the C447 of human ERα is C451. The physiological roles of the palmitoylation of C451 were tested in mouse models with a point mutation at C451 to non-palmitoylable alanine (C451A) [73,74]. This mutation leads to female infertility [73,74]. Detailed reproduction phenotyping revealed that the C451A ERα mutant female mice can be pregnant but show a series of reproduction-associated abnormalities such as embryonic/neonatal death, delayed parturition, and small litter size [89]. This study showed abnormal placental development including altered behavior of a trophoblast cell population, which supports placentation and nourishes embryos. Excessive progesterone levels were also observed in this study, which may be responsible for the delayed parturition because progesterone withdrawal is required for uterine contraction to initiate labor. Taken together, these observations indicate that membrane-bound ERα by palmitoylation on C451 is important to support female reproduction, from an endocrine mechanism and placenta development. Although membrane-initiated ERα signaling is involved in estrogen-mediated energy homeostasis [90,91,92], C451A ERα mutant mice do not show body weight phenotype and abnormal fat accumulation [74,75,89]. E_2_ prevents bone demineralization, which is significantly impaired in the mutant mice, indicating that this effect is primarily exerted by membrane-bound ERα [75].

#### 3.2.5. Y537 Phosphorylation

Tyrosine 537 (Y537) is located in the ERα LBD/AF-2 domain. Phosphorylation of Y537 regulates E_2_ binding and the receptor dimerization in MCF-7 cells, and the interaction with kinases containing SH2 domains such as Src and HER-2 [76,77]. Y537N mutations are identified in tumors of metastatic cancer patients, and this mutation leads to the constitutive transactivation activity [93].

The mouse counterpart of the Y537 human ERα is Y541. Simond et al. generated a mouse model with a Y-to-serine (S) point mutation at 541 residue causing constitutive activation of ERα [78]. The mutant mice showed tissue-specific developmental defects. The mutant females are infertile. Their uteri show abnormal distribution of epithelial cells, and a greater number of follicles are found in their ovaries. Male mutants develop atrophy of the testes and seminal vesicles. Of note, the mutant males are feminized in their nipples and anogenital region as indistinguishable from female mutants. Both female and male mutants display a significant development of bone area.

### 3.3. New Unidentified Variant

Recently, we found the presence of a new ERα variant that has a particular physiological role in mice [36]. In this study, a lox-flanked transcription blocking cassette was introduced right before the first coding exon of the ERα gene, which was expected to cause the knockout of ERα expression and restore ERα expression in the presence of Cre recombinase. Unfortunately, this genomic modification did not completely eliminate ERα protein expression. Instead, it resulted in biased knockdown of a particular ERα variant. ERα was detected as a doublet around 66 kDa in WT mice tissues in WB. Our genomic modification remarkably reduced only the upper band of the ERα66 doublet. The upper band was predominant in the pituitary, testis, and ovary, but minimally expressed in the brain, where the lower band was dominant. We tentatively reported this upper band as ERα66* in contrast to ERα66. The ERα migration pattern in our WB study somewhat resembled the phosphorylated ERα66 observed in previous studies. Korach et al. have reported that the upper band of the doublet is strongly induced in the presence of estradiol, preferentially extracted from the nuclear fraction, and more heavily phosphorylated in mouse uteri [94,95]. Joel et al. have reported that 66-kDa ERα was upshifted by single phosphorylation of S118 of human ERα, and downshifted by phosphatase treatment when it was transfected in monkey kidney-derived COS cells [96]. Tabatadze et al. have observed an ERα doublet around 66 kDa in rat brains, the upper band of which is a phosphorylated ERα preferentially localized in synaptic vesicle fractions [97]. A phosphorylation-dependent ERα doublet at 66 kDa is also reported in human tissues and cells [98,99]. We tested if the generation of the ERα doublet was due to ERα phosphorylation, and showed that the dephosphorylation treatment did not change the upper band [36], suggesting that the generation of the doublet is not likely due to ERα phosphorylation. Rat pituitary full-length ERα and ERαΔE3 in a previous study [28] are similar to the ERα migration pattern observed in our WB study. If we extrapolate their finding to our observation, full-length and ERαΔE3 would correspond to ERα66* and ERα66, respectively. Despite our efforts to determine the transcript sequence and N-terminal peptide sequence, we have failed to identify its structure thus far. Although the identity of this variant remains undetermined, it is unlikely that a genetic modification made in the noncoding region of a gene influences the PTMs of its protein product. Thus, based on the transgenic strategy used in this novel mouse model, we concluded that this upper band might represent a novel ERα splicing isoform that has not been appreciated previously.

Importantly, our transgenic mice with selective downregulation of ERα66* isoform caused biased physiological consequences [36]. While this mutant mouse did not show obvious metabolic phenotypes compared to WT mice, female reproductive or maternal functions were impaired; female mutants were capable of pregnancy despite their irregular estrus cycle, but viable newborns were rarely found. It is essential to determine the identity and the regulatory mechanism of the expression of ERα66* for a better understanding of its tissue-specific physiological functions.

## 4. Perspectives

We have discussed the physiological functions of ERα splice variants and PTMs independently. PTMs of the splice variants remain largely unexplored. Considering that some splicing variants and PTMs share cellular and physiological functions, it is not surprising that PTMs of ERα splicing variants impact their functions. In fact, ERα36 is phosphorylated by IKKε at Serine 167 [100]. Stirone et al. found multiple isoforms of ERα in rat cerebral blood vessels at 110, 93, 82, 50, 45 kDa as well as at 66 kDa for the putative receptor; and 82-kDa ERα was detected by N-terminal, C-terminal, and phosphorylated S118-specific antibodies [47]. Since it is not realistic that single phosphorylation would cause this massive upshift of WB bands, this 82-kDa ERα phosphorylated at S118 would be a phosphorylated ERα splice variant or phosphorylated ERα heavily undergoing other PTMs. Diverse subcellular distribution of ERα46 may reflect differential PTMs on this isoform [11,12,13]. The development of epitope-specific antibodies recognizing the PTMs of ERα has greatly helped the analysis of a certain PTM in various conditions. Unfortunately, however, most of the studies only focused on PTMs of full-length ERα and provided WB images around 66 kDa. An association study of splicing isoforms and PTMs will be helpful to understand the PTM status of alternative splice variants even if their identities are not yet determined.

A revolution of RNA sequencing technology has revealed an unexpectedly wide range of splice events, and how dynamically alternative splicing is changed in different physiological and pathological contexts [101,102]. However, very little is known about how alternative splice isoforms functionally impact cellular or in vivo phenotypes. A limitation comes from the lack of tools to properly study the role of a particular splice isoform. In vivo study for physiological functions of alternative splicing variants without affecting the endogenous expression of canonical protein is extremely difficult because they share the same exons. An RNA-directed nucleic acid editing technology, CRISPR/Cas13, was shown to efficiently target RNA instead of DNA in CRISPR/Cas9 [103,104]. CRISPR/Cas13 may enable the targeting of a particular splice variant of a gene in a cell type-specific manner to study the physiological functions of individual splice variants.

Through the extensive functional studies for ERα since its cloning, the structure of the ERα protein has been subdivided into six functional domains (A to F domains). However, a significant part of ERα is intrinsically disordered, where a protein does not have a rigid 3D structure and the functionalities seem to be context-dependent. The majority of transcription factors possess intrinsically disordered protein (IDP) domains, which are more likely to undergo PTMs and alternative splicing than structured domains [105]. ERα is also intrinsically disordered, especially in the N-terminal domain [106,107], and many of the ERα PTMs and splicing sites appear to be located at IDP domains [56,57]. The coordination of alternative splicing and PTMs in IDP domains is proposed to provide cells and organisms with flexibility in protein functionality in a context-dependent manner. Much more effort is needed to replenish our knowledge on the splice variants and PTMs of ERα that have mostly been studied independently.

## Figures and Tables

**Table 1 cells-12-00895-t001:** ERα alternative splicing variants and their physiological functions and/or properties.

Isoform	MW (kDa)	Species	Tissues Detected	Physiological Functions/Properties	References
ERα46	46	Human	ovary, liver, skeletal muscle, lung, kidney, adrenal gland	Inhibition of ERα66-dependent transactivation Mostly localize in nucleus while the significant portion is also associated with plasma membrane in transfected cells	[9,10,11,12]
		Mouse	heart, uterus, testis	Localize primarily to the membrane fraction	[13]
		Rat	pituitary	Protein expression is found in the pituitary	[12]
ERα36	36	Human	breast cancer endometrial cells, mammary gland, uterus	Inhibition of ERα66-dependent transactivation	[14,15]
				Overexpression of human ERα36 in mouse mammary gland leads to constitutive estrogenic activation phenotypes	[16]
				Overexpression of human ERα36 changes the transcription of genes in cell proliferation, meiotic division, and differentiation in male germ cells	[17]
				Localize to plasma membrane	[13,18]
		Mouse	kidney, ovary, oocytes	Alternative terminator exon of human ERα36 does not exist in mice. However, the immunoreactivity against ERα36-specific antibody is detected in mouse	[13]
ERαV/ΔE3-6	37	Human	brain, MCF-7 cells	The expression is increased by NPE3-3 overexpression MCF-7 cells while ERα66 is reduced Inhibit E_2_-initiated transcriptional activity of ERα66 in Hela cells and neuroblastoma cells. The expression is increased by E_2_ in cell type-specifically No physiological study available	[19,20]
ERα-LBD	37.3	Human	breast cancer	The overexpression facilitates cancer growth and develops the resistance to the ERα antagonist Fulvestrant Predominantly localized to the cytoplasm and mitochondria No physiological study available	[21]
NTERPs	37.3	Mouse	pituitary, brain, ovary, testis, kidney	Localized to extra-nucleus Lacks E_2_-dependent transcriptional activity No physiological study available	[22]
TERPs	20–24	Rat	pituitary	The expressions have been reported in rodents but not in humans	[22,23,24,25,26]
				The transfection in CHO cells inhibits ERE-binding of ERα, E_2_ binding to ERα, E_2_-induced transcriptional activation	[24]
ERαΔE3/ΔE4/ΔE3/4	61.8/53/45	Rat	pituitary	Expression is developmentally regulated ΔE3; inhibition of E_2_-dependent ERα transactivation ΔE4; induction of ligand-independent transactivation	[27,28,29,30]
ERαΔE4	52	Human	MCF-7 cells, pituitary, ovarian carcinoma	The transcript has been reported in various tissue while the protein expression has been reported in ovarian carcinoma	[31,32,33]
	52	Mouse	N-38 immortalized hypothalamic neurons	Localized to plasma membrane	[30,34,35]
ERα66*	67–70	Mouse	pituitary, brain, ovary, testis, liver	Appears as the upper band of a doublet at 66 kDa Female reproduction/maternal behavior	[36]

## Data Availability

Not applicable.

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
