# Peer review of "Estrogen Receptor Alpha Splice Variants, Post-Translational Modifications, and Their Physiological Functions"

_cells, 2023, doi:10.3390/cells12060895_

Round 1
Reviewer 1 Report
This review from Saito and Cui focuses on physiological functions of estrogen receptor alpha splice variants.
The manuscript is well articulated and takes into account most of the literature concerning this topic.
Since functional relationship and stechiometric expression balance between ERα full length and splice variants might have a clinical significance in receptor positive cancer management, I wonder why the authors did not consider this point at all. Indeed, I suggest to add a paragraph including the correlation of ERα splice variants with response to cancer therapy and resistance, so as to increase its interest from a more translational and clinical point of view.
Author Response
- This review from Saito and Cui focuses on physiological functions of estrogen receptor alpha splice variants.
- The manuscript is well articulated and takes into account most of the literature concerning this topic.
- Since functional relationship and stechiometric expression balance between ERα full length and splice variants might have a clinical significance in receptor positive cancer management, I wonder why the authors did not consider this point at all. Indeed, I suggest to add a paragraph including the correlation of ERα splice variants with response to cancer therapy and resistance, so as to increase its interest from a more translational and clinical point of view.
Response:
We appreciate that we appear to share same awareness on the clinical relevance of ERα splicing variants.
As you pointed out, the expression of ERα splicing variants and a certain PTM of it have been extensively studied especially in cancer research field. Even though some ERα splicing variants and their PTMs are reported in normal physiological conditions, however, little attention is paid on physiological functions of them in normal conditions. The focuses of the review actually are to fill gap of our current knowledge and to emphasize the importance to study in functions of ERα variants and PTMs in normal physiological conditions.
We want to respectfully point out there are excellent reviews on ERα variants in cancer research field. We referred the readers to some of them in the main text. As in the main text, we cite following four review articles.
- Herynk, M. H. and S. A. Fuqua. "Estrogen receptor mutations in human disease." Endocr Rev 25 (2004): 869-98.
- Taylor, S. E., P. L. Martin-Hirsch and F. L. Martin. "Oestrogen receptor splice variants in the pathogenesis of disease." Cancer Lett 288 (2010): 133-48.
- Gong, Z., S. Yang, M. Wei, A. C. Vlantis, J. Y. K. Chan, C. A. van Hasselt, D. Li, X. Zeng, L. Xue, M. C. F. Tong, et al. "The isoforms of estrogen receptor alpha and beta in thyroid cancer." Front Oncol 12 (2022): 916804.
- Thomas, C. and J. A. Gustafsson. "The different roles of er subtypes in cancer biology and therapy." Nat Rev Cancer 11 (2011): 597-608.
Reviewer 2 Report
The paper entitled "Estrogen receptor alpha splice variants, post-translational modifications, and their physiological functions" delves into the different splice variants and post-translational modifications of ER-alpha.
The review is well written, interesting and provides recent citations.
However, there are certain aspects that I would like to emphasize:
The authors mention in the title and abstract of the paper that they are going to list different ER-alpha variants and post-translational modifications associating them with a physiological process. However, the physiological process is not mentioned in the manuscript, especially when they list the ER-alpha splice variants.
They refer to many studies performed on the MCF-7 breast cancer cell line and not studies that observe these variants in their normal cellular processes.
Therefore, some physiological processes of the ER variants should be introduced or the title and objective of the review should be changed, as little emphasis is placed on the true physiological functions of these variants and it is mainly molecular data of these variants.
Author Response
- The paper entitled "Estrogen receptor alpha splice variants, post-translational modifications, and their physiological functions" delves into the different splice variants and post-translational modifications of ER-alpha.
- The review is well written, interesting and provides recent citations.
Response:
We appreciate the reviewer's positive compliment.
- However, there are certain aspects that I would like to emphasize:
- The authors mention in the title and abstract of the paper that they are going to list different ER-alpha variants and post-translational modifications associating them with a physiological process. However, the physiological process is not mentioned in the manuscript, especially when they list the ER-alpha splice variants.
- They refer to many studies performed on the MCF-7 breast cancer cell line and not studies that observe these variants in their normal cellular processes.
- Therefore, some physiological processes of the ER variants should be introduced or the title and objective of the review should be changed, as little emphasis is placed on the true physiological functions of these variants and it is mainly molecular data of these variants.
Response:
Limited knowledge of their true physiological functions exactly is a motive of current review. It is very difficult to tease out variant-specific physiological functions because full-length ERα and splice variants share exons each other and disruption of any exons could also affect other variants. As a result, with a few exception (Thiebaut's ERα36 overexpression models and our ERα hypomorphic mice), most of splicing variant part is occupied by their molecular data including domain structures, expression sites, and physiological functions at cellular level. We have discussed the limited studies of the variant-specific physiological functions in second paragraph of "Perspective" chapter (Line 420) and added a sentence to note the limited description about the variant-specific physiological functions in the introductory paragraph of third chapter "ERα isoform variation and functions".
Reviewer 3 Report
The manuscript of Saito and Cui is an excellent review on estrogen receptor alpha (ERα) splice variants, post-translational modifications, and their physiology. Several ERα splicing variants have been described without corresponding understanding of the physiological functions of each of those isoforms. Alternative splicing is an important molecular mechanism that enables the production of a variety of different transcripts from a single gene by selectively arranging coding exons from pre-mature mRNA. Many of them are reported particularly in cancers and tumor cells, but some of them are also expressed constitutively in normal tissues. In this review, the authors focus on identified functional diversity of ERα variants in physiological conditions. In addition, they provide a detailed description of post-translational modifications (phosphorylation, methylation, and palmitoylation) and their physiological relevance.
This is very well written review on an important subject. Figure 1 and the table 1 are perfect presentations of ERα gene organization and the physiological functions of ERα splicing variants, respectively. The citation of contributors to the field is detailed and scholarly.
Author Response
- The manuscript of Saito and Cui is an excellent review on estrogen receptor alpha (ERα) splice variants, post-translational modifications, and their physiology. Several ERα splicing variants have been described without corresponding understanding of the physiological functions of each of those isoforms. Alternative splicing is an important molecular mechanism that enables the production of a variety of different transcripts from a single gene by selectively arranging coding exons from pre-mature mRNA. Many of them are reported particularly in cancers and tumor cells, but some of them are also expressed constitutively in normal tissues. In this review, the authors focus on identified functional diversity of ERα variants in physiological conditions. In addition, they provide a detailed description of post-translational modifications (phosphorylation, methylation, and palmitoylation) and their physiological relevance.
- This is very well written review on an important subject. Figure 1 and the table 1 are perfect presentations of ERα gene organization and the physiological functions of ERα splicing variants, respectively. The citation of contributors to the field is detailed and scholarly.
Response:
We appreciate that our manuscript was taken as we intended. Although we did not make any changes in response to your comments, since we made some changes in response to other reviewers, we hope our revised edition does not change your opinion from the original submission.
Reviewer 4 Report
This is a well-written comprehensive review on the physiological functions of estrogen receptor alpha splice variants and its posttranslational modifications.
Minor points:
1. In Fig. 1, in addition to the shown mouse ERa splice variants, are the additional variants indicated in Table 1 as human variants (like ERαV/ΔE3-6) also expressed in mice?
2. In Table 1, the "Physiological functions" indicated for each variant should be clearer identified regarding their source of data (at least the species and tissue).
3. Have all splice variants shown in Table 1 been corroborated at the protein level? If not, it should be indicated, which variants were detected at the mRNA level only.
4. An additional figure illustrating the different functions of the splice variants and PTMs addressed in this article would be helpful for the reader.
Author Response
- This is a well-written comprehensive review on the physiological functions of estrogen receptor alpha splice variants and its posttranslational modifications.
Response: We appreciate the reviewer's compliment.
- 1. In Fig. 1, in addition to the shown mouse ERa splice variants, are the additional variants indicated in Table 1 as human variants (like ERαV/ΔE3-6) also expressed in mice?
Response: We thank you for pointing this out. We missed putting an annotation on ERαV/ΔE3-6, which has been corrected by putting superscript "a" indicating that is not fully characterized in rodents. Figure. 1 summarizes mouse ERα variants, whose human counterparts are also expressed in human. The exceptions are ERα36,ERαV/ΔE3-6, and ERα-LBD, the expressions of which are not yet fully proven in rodents. In addition, we did our best to clarify if both rodents and human express the corresponding ERα splicing variants in the main text. We hope our intention will be taken well by the readers.
- 2. In Table 1, the "Physiological functions" indicated for each variant should be clearer identified regarding their source of data (at least the species and tissue).
Response: We re-organized Table 1 as to correspond described physiological functions to their original articles.
- 3. Have all splice variants shown in Table 1 been corroborated at the protein level? If not, it should be indicated, which variants were detected at the mRNA level only.
Response: This comment prompted us to re-check the protein expression proofs of described variants and let us realized that we did not mention human counterpart of ERαΔ4 protein expression. This was corrected in revised Table 1. Now ERα variants in Table 1 are described at both transcript and protein levels in the original articles. Precisely speaking, no study here provided protein sequencing results. However, at least, the reasonable validations of protein expressions were provided for all the splicing variants mentioned here. For example, many studies performed transfections of the corresponding transcript construct followed by SDS-PAGE/WB and showed protein bands at expected molecular weights. Accordingly we added the following sentences in the end of the paragraph 3.1.6 at the line 220.
"ERαΔ4 has been reported in immortalized mouse hypothalamic neurons [30, 34], and the mouse brain [35] and predominantly located in the cell membrane [29]. While human ERαΔ4 transcript is reported in various cell lines and tissues including MCF-7 cells [31], human pituitary [32], the protein expression is confirmed in ovarian carcinoma [33]."
- 4. An additional figure illustrating the different functions of the splice variants and PTMs addressed in this article would be helpful for the reader.
Response: This would be certainly helpful. Since the integration of both the splice variants and the PTMs in one figure will be busy, however, we added Table 2 summarizing the findings in PTM mouse models that we mention in the main text (Line 267).